



# Using PhET™ Interactive Simulation Plate Tectonics on Initial Teacher Education

Bento Cavadas[1], Sara Aboim[2]

[1]Department of Mathematics and Science, Polytechnic Institute of Santarém/School of Education, Santarém, 2001-902 Santarém, Portugal;
CeiED – Interdisciplinary Research Centre for Education and Development, University Lusófona, Portugal.
[2]Department of Mathematics, Science and Technologies, Polytechnic Institute of Porto/School of Education, Porto, 4200-465 Porto, Portugal
inED - Centre for Research and Innovation in Education, Polytechnic Institute of Porto/School of Education, Portugal

*Correspondence to*: Bento Cavadas (bento.cavadas@ese.ipsantarem.pt)

**Abstract.** Using digital educational resources in science education is an effective way of promoting students content knowledge of complex natural processes. This work presents the usage of a digital educational resource (DER) CreativeLab_Sci&Math | Plate Tectonics, that explores the PhET™ Plate Tectonics simulator, in the context of preservice teachers (PST) education in Portugal. The preservice teachers performance was analysed on the five tasks in which the DER was organized. Results show that the DER contributed for the successful achievement of PST of the following learning outcomes: describing the differences between the oceanic crust and continental crust regarding temperature, density, composition and thickness; associating the plate tectonic movements with their geological consequences; identifying the plate tectonic movements that cause the formation of some geological structures. Results also show that PST considered PhET™ Plate tectonics simulator contributed for their learning about plate tectonics.

## 1 Introduction

This work presents the implementation of the digital educational resource (DER) CreativeLab_Sci&Math | Plate Tectonics elaborated under the CreativeLab_Sci&Math project (Cavadas, Correia, Mestrinho and Santos, 2019). This resource is a structure guide with several tasks that explores the simulator PhET™ Plate Tectonics from the PhET Interactive Simulations™ of the University of Colorado Boulder. This simulator aims to explore the plate tectonic dynamics on Earth´s surface through the manipulation of different variables, such as temperature and density, and crust types (oceanic crust and continental crust). The tasks of the educational resource were organized according to the following learning outcomes (Lo):

- Lo1: Describe the differences between the oceanic crust and the continental crust regarding temperature, density, composition and thickness;
- Lo2: Associate the tectonic plate movements with their geological consequences;
- Lo3: Identify the tectonic plate movements that cause the formation of certain geological structures.



This educational resource was applied to the Initial Teacher Education (ITE) in two Portuguese Teacher Training Institutions (TTI). This work presents the results of its implementation in the academic year of 2019/20. The main question that guided the work was: "What was the contribution of the educational resource CreativeLab_Sci&Math | Plate tectonics to the preservice
35  teachers´ learning about plate tectonics?"

## 2 Learning and teaching about plate tectonics

In the framework of history of science, the explanations for the earth's surface origin, evolved from the contracting earth to the Continental Drift theory (Wegener, 1966) and, later, to the more unifying and integrative theory of Plate Tectonics (Frankel 2012a, 2012b, 2012c). The Plate Tectonics theory was supported by evidences associated to the seafloor spreading, seafloor
40  sediments, magnetic anomalies, and the geological meaning of mid-ocean ridges, rifts, oceanic trenches and transform faults. Studies about convection, the formation of mountain ranges, Benioff zone, the subduction process, the tectonic plate boundaries and their characteristics also contributed for the construction of the Plate Tectonics theory (Wilson, 1966; Frankel, 2012c).

That knowledge about Earth dynamics suffered didactic transposition into science textbooks (Cavadas, 2019) and promoted
45  reflections about Plate Tectonics teaching. In fact, a few years after the enunciation of Plate Tectonics theory by Morgan, in 1967 (Frankel, 2012c), studies about its didactic transposition for educational contexts appeared, such as the work of Glenn (1977). This researcher presents suggestion for teaching elementary school students about Continental Drift and Plate Tectonics. In Portugal, other works were published in the following decades about: the didactic transposition of Continental Drift and Plate Tectonics into textbooks (Cavadas 2019; Cavadas and Franco, 2009; Faustino, et al. 2017; Santos, et al. 2010);
50  the teaching of those subjects in an history of science and epistemological perspective (Almeida, 2000; Praia, 1995; Vasconcelos, et al., 2013); and specific topics related to Plate Tectonics, such as palaeomagnetism (Correia, 2014).

Students' misconceptions about Plate Tectonics were also comprehensively studied (Borges, 2002; Dolphin and Benoit, 2016; Francek, 2013; Mills et al., 2017). Francek (2013), in a global study about 500 hundred misconceptions on geosciences, concluded that about 19% was related with Plate Tectonics. Regarding this matter, Marques and Thompson (1997), in a sample
55  of 16/17 year old Portuguese students, identified misconceptions about plates and their motions, such as: plates are arranged like a stack of layers (64%), the same tectonic plate mechanism causes continental and oceanic mountain ranges (40%) or magnetic polar wandering causes the motion of plates (34%). In a more recent study, Mills, Tomas and Leuthwaite (2017) concluded that many 14 year old students (n=95) also had misconceptions about the nature, movement, boundaries of tectonics plates and the occurrence of geological events at tectonic plate boundaries. For example, many students thought tectonic plates
60  were located underground and they were not exposed on the Earth's surface; Earth's spin axis causes tectonic plates to move; tectonic plate boundaries are located at the edge of continents or countries; and that earthquakes are caused when two tectonic





plates suddenly crash together. The identification of these misconceptions about plate tectonics, many of them repeated in different studies, is a clear evidence of the persistency of those misconceptions among students.

According to Marques and Thompson (1997), traditional Earth science teaching methodologies in Portuguese schools do not
contribute for the eradication of those student' misconceptions. One alternative approach is using the potentialities of digital resources to teach plate tectonics. Some studies found learning about some contents of plate tectonics using digital resources as Google Earth® (Bitting et al., 2019; Ferreira, 2016) is beneficial for students, for example, in the learning of mountain range and volcanos formation and the distribution of earthquakes (Bitting et al., 2019). Mills et al. (2019) showed the use of student-constructed animation by 11-14 year old students, for explaining the processes that occur in tectonic plate boundaries,
contributed for their better performance in following GeoQuiz about those processes. Therefore, Mills et al. (2019) concluded that work proposals based on representations of plate tectonic processes contribute for students' learning about those processes.

## 3 Interactive simulations and science learning

Computer simulations have undergone a great evolution. In their beginning, they were simple models, in which interactivity was possible with only one or two defined parameters. Currently, simulations are more complex, with more realistic visual
representations which allow the user to make changes and observe their effects in real time. In addition, the simulations have unique characteristics which are not present in many other learning tools, such as, interactivity, animation, dynamic feedback, exploration and discovery (Podolefsky et al., 2010).

One of the most effective ways to solve a problem is by simulating reality, replicating one specific situation for better analysis and study (Tan, 2007). The use of digital simulations is very important in Earth sciences since it allows the study of certain
processes that cannot be reproduced in laboratory and, therefore, the exploration of the relations between the theoretical framework and the simulated geological processes observed. Simultaneously, it improves the motivation and interest of students in classes (Nafidi et al., 2018; Quintana et al., 2004; Pinto et al., 2014). Digital simulations can be used as educational resources to promote observation, communication, analysis, hypothesis formulation and critical thinking skills of students (Nafidi et al., 2018). Problem solving and scientific discovery learning in digital simulations allow students to build new and
meaningful knowledge about what they are learning (de Jong and Joolingen, 1998) and reflecting about their learning, in a metacognition process (Droui, 2014; Nafidi and Hajjami, 2018).

Other studies, in the framework of teacher education, revealed benefits from the use of simulations (Trundle and Bell, 2010) and multimodal digital animations, known as Slowmotion, for science learning (Hoban, et al., 2011; Paige, et al. 2016), validating the use of digital education resources in teacher science education.

Regarding PhET™ simulations, that were used in the present study, some works showed they are very useful for engaging students and improving their interest and knowledge in many scientific fields (Hensberry et al., 2013; Lancaster et al. 2013; McKagan et al., 2008; Perkins et al. 2012; Wieman et al. 2010). PhET™ simulations are created to promote science education,





are freely available on the PhET™ website and they are more effective for conceptual understanding (Podolefsky et al., 2010; Perkins et al., 2012). Simulations are designed with little text information, so that students can easily use it in the classroom,

in a laboratory or as homework (Podolefsky et al., 2010). Since PhET™ simulations are digital resources available online, they can be used and explored in the distance and online learning modality, as recommended by Commonwealth of Learning, in the statement *Keeping the doors of learning open COVID-19* (COL, 2020).

The current work is innovative because it is focused on the use of a simulator, PhET™ Plate Tectonics, in teacher training context about Plate tectonics. There is also innovation concerning the creation and the implementation of the DER resource

CreativeLab_Sci&Math Plate tectonics, because it resulted from the collaboration between teachers of two different Schools of Education, enabling the exchange of experiences and practices between institutions that are involved in teacher education, including online learning experiences.

## 4 Methods

### 4.1 Participants

Participants on this study were 68 preservice teachers (PST) of two Portuguese TTI's, ranging from 19 to 38 years old. Those PST at the end of their graduation can teach children from kindergarten to the initial years of elementary school, but not high school students. The high school background of the majority of PST is linguistics. Only a few of them attended science courses in high school before achieving Higher Education.

The study followed the guidelines and recommendations of the authors' research centres ethical committees. All participants

authorized the use of their data and written productions for science education research purpose, through informed voluntary consent. Participants were clearly informed that they could withdraw from the investigation at any time and that their data would be anonymized during data analysis.

### 4.2 Design and implementation of the educational resource

Before the current implementation, the educational resource CreativeLab_Sci&Math | Plate Tectonics was implemented during

the two preceding academic years, in the context of ITE of one Portuguese TTI. During that implementation, the educational resource was constantly improved concerning its didactic sequence, task's approach and use of the simulator's potentialities, following PST' feedback and teacher's reflections. It was also peer-reviewed by another TTI science education teacher.

The final version of the resource was organized in tasks in a GForm®. The GForm® has the advantage of allowing the inclusion of different types of questions (multiple choice, checkboxes, etc.) and resources (text, images, videos, links) and the benefit of

giving immediate feedback to the students about their performance in each task and globally. The educational approach used, was guided Inquiry because it is the didactic approach recommended for the use of PhET™ simulations (PhET, 2014).



Accordingly, in the design and implementation of the educational resource, the following didactical recommendations of PhET™ simulations were considered:

- Specific learning outcomes were defined (e.g. Lo1, Lo2 and Lo3);
- Minimal instructions for the use of the simulator were delivered, as it was developed with the aim of allowing free exploration by PST, whose role is to construct a useful meaning from those explorations;
- PST were stimulated to mobilize their previous knowledge about Plate tectonics in the tasks;
- PST were encouraged to use their problem-solving skills to give correct answers to the problems posed in the tasks;
- Tasks were structured to be performed in pairs of PST, encouraging cooperation and discussion of ideas;
- A reflection about the contribution of the educational resource to their learning was proposed to PST. It was also asked if they had any suggestions addressing the improvement of the educational resource.

Table 1 shows the relationship between the tasks and the learning outcomes.

**Table 1: Relationship between the educational resource CreativeLab_Sci&Math | Plate Tectonics tasks and the learning outcomes.**

| Tasks | Learning outcome |
|---|---|
| Task A1 and A2. Characteristics of crust | Lo1: Describe the differences between the oceanic crust and the continental crust regarding temperature, density, composition and thickness. |
| Tasks B1 and B2. Plate movements | Lo2: Associate the tectonic plate movements with its geological consequences. |
| Task C. Inquiry about plate tectonics | Lo3: Identify the tectonic plate movements that cause the formation of some geological structures. |


The resource was implemented in classes of two different Portuguese TTI's. In one of them, the educational resource was implemented in Earth and Life Sciences curricular unit, and in the other one, in Geosciences curricular unit. The tasks were performed in pairs of PST. This had the advantage of promoting discussion between them. The timeframe to accomplish the tasks was two hours.

**4.3 Methods of data collection**

Several types of data were collected for this study. One was the PST´ answers to the tasks of the educational resource. Another one was PST´ reflections concerning the contribution of the educational resource to their learning, as also the suggestions for its improvement, through a short survey. Research teachers course materials and reflections about the observation of PST' work were also collected.



## 4.4 Data analysis

In the results and discussion section it is presented a detailed description of each task. PST' performance was analysed for each question of tasks A, B and C. The correct answers for each question were quantified, the relative frequency was calculated and possible justifications for student's achievement were presented.

At the end of the tasks, a sample of students (19 pairs) was asked to give feedback about the contribution of the educational resource to their learning of plate tectonics, and also if they had any suggestions addressing the improvement of the resource. Through a post-categorization of PST' answers, a qualitative analysis of these data was made. Two main categories of analysis were defined: "Contributions to learning" and "Improvement suggestions". Extracts of PST´ answers were used to better support the analysis.

## 5 Results and discussion

In this section are presented the results of the simulations application for each task and a discussion of PST performance.

## 5.1 Task A. Characteristics of crust

For the study of crust characteristics, in task A1, PST had to explore different crust conditions, such as density, temperature and thickness, and classify three statements (A1.1., A1.2. and A1.3.) as "true" or "false". Another purpose of A1 tasks was to give PST a first approach to the simulator sections and toolbox and engage them in the following tasks.

Regarding statement A1.1. "Oceanic crust is denser than continental crust", PST should use the density meter in the toolbox to compare both crusts´ density, as shown in figure 1. That exercise will allow them to perceive that oceanic crust density is higher than the continental crust.

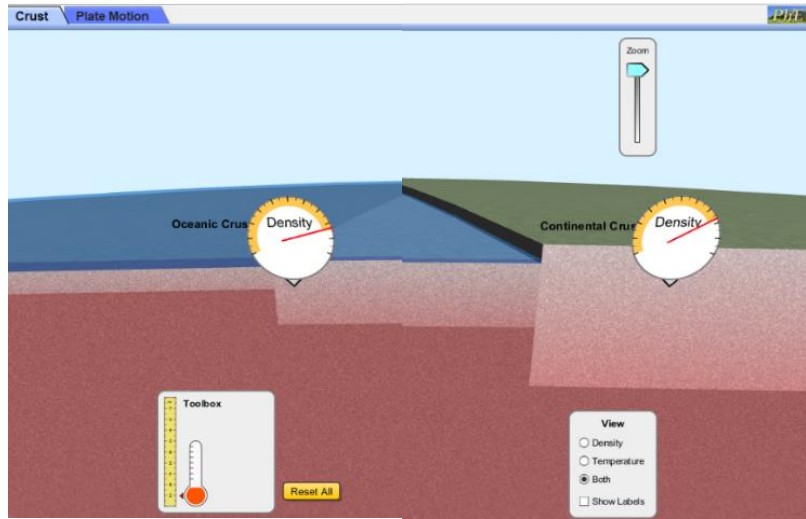

**Figure 1: Density values of oceanic and continental crust (Caption from PhET™ Plate Tectonics).**





Concerning statement A1.2., "The temperature of oceanic crust is higher than continental crust at 5km depth", PST were expected to use the thermometer and the rule in the toolbox. By positioning the thermometer at the depth of 5km in oceanic and continental crust, PST will observe that the temperature of oceanic crust is slightly higher than continental crust, as shown in figure 2.

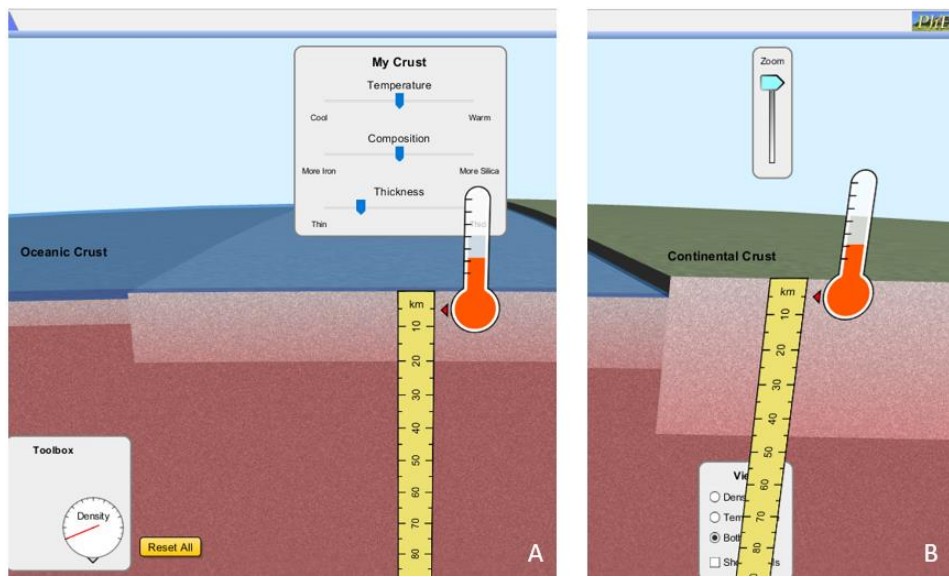


**Figure 2: Oceanic (A) and continental crust (B) temperature at 5 km depth (Caption from PhET™ Plate Tectonics).**

The next statement A1.3. was related with crust thickness: "Oceanic crust is thicker than continental crust." Using the rule in the toolbox, PST could perceive that this statement is false because continental crust is thicker than oceanic crust.

In task A2, PST had to analyse vertical movements of the crust by changing some variables. In task A2.1., they were expected 175 to analyse what happens to the crust when its temperature increases, maintaining the same density and thickness in the panel "My crust". By doing so, they should verify that the crust moves slightly upward. By decreasing the temperature, they should observe the opposite. After that, they should select one of two possible answers, being the following answer the correct one: "As the temperature of crust increases, the density of the materials that compose it decreases. The less dense crust rises as a result of the denser composition of the mantle"

Task A2.2. concerned the density of the crust. Through changing crust' silica and iron composition in "My crust", maintaining the same temperature and thickness (Figure 3), PST should conclude that the following answer is the correct one: "There is a direct relationship between the percentage of iron in the composition of the crust and its increased density. The denser crust descends as opposed to the less dense composition of the mantle.



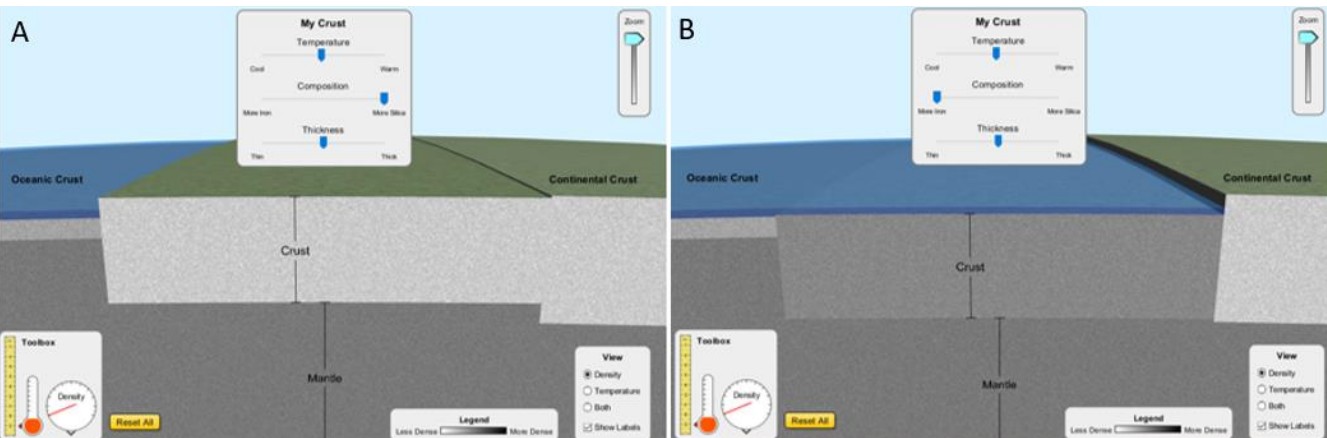

**Figure 3: Comparison between the movements of a crust that is richer in silica (A) and a crust that is richer in iron (B) (Caption from PhET™ Plate Tectonics).**

Table 2 shows PST tasks A1.1. to A2.2 results.

**Table 2. Preservice teachers' tasks A1.1. to A2.2 results.**

| Task | Frequency of correct answers (n=68) |
|---|---|
| A1.1. Comparison of oceanic and continental crust density. Correct answer: "Oceanic crust is denser than continental crust." | 80,9% |
| A1.2. Comparison of oceanic and continental crust temperature. Correct answer: "The temperature of oceanic crust is higher than continental crust at 5km depth." | 76,5% |
| A1.3. Comparison of oceanic and continental crust thickness. Correct answer: "Oceanic crust is thicker than continental crust" | 100% |
| A2.1. Select the process that occurs when crust's temperature increases. Correct answer: "As the temperature of the crust increases, the density of the materials that compose it decreases. The less dense crust rises as a result of the more dense composition of the mantle" | 83,8% |
| A2.2. Select the process that occurs when crust's density increases. Correct answer: "There is a direct relationship between the percentage of iron in the composition of the crust and its increased density. The more dense crust descends as opposed to the less dense composition of the mantle" | 88,2% |


Results show that, using the simulator, PST performed very well on A1 tasks. The performance results on task A1.2 were relatively lower., in comparison to the other A1 tasks. This could be due to the smaller differences on both crusts´ temperature



that are difficult to observer on the simulator thermometer´s scale, at 5 km depth. On the other hand, a small mistake on the positioning of the thermometer at 5 km depth could also cause an incorrect reading of the temperature.

As in task A1, PST score in task A2 was also very high. The observation of the crust movements in the simulator was very helpful to identify the influence of crust's temperature and composition on its movements, in relation to the Earth's mantle. Some incorrect answers on A2.1. may have resulted from the fact that the crust movement, due to temperature variation, is relatively small and sometimes difficult to observe in the simulator. On A2.2. some incorrect answers may have resulted from a poor conceptual knowledge of the meaning of density, since many PST had a poor background on geosciences. However, the good results on tasks 2.1. and 2.2. could contribute to avoid the common misconception that "Vertical forces push up the bottom of the oceans and originate the continents." (Marques and Thompson, 2006, p. 207).

### 5.2 Task B. Plate movements

Task B was elaborated with the purpose of studying the tectonic plate movements (Figure 4). Initially, PST must select the section "Plate movements" and reproduce the tectonics situations presented trough captions of PhET™ Plate Tectonics in each task.

The initial tasks (B1) regarded the analysis of processes that occur on convergent plate boundaries. In task B1.1., PST should examine the collision between a plate with continental crust and a plate with old oceanic crust (old oceanic plate). Four options for an answer were given, and through playing the simulation, PST should notice that the correct one was the answer stating that the oceanic crust suffers subduction under the continental crust.

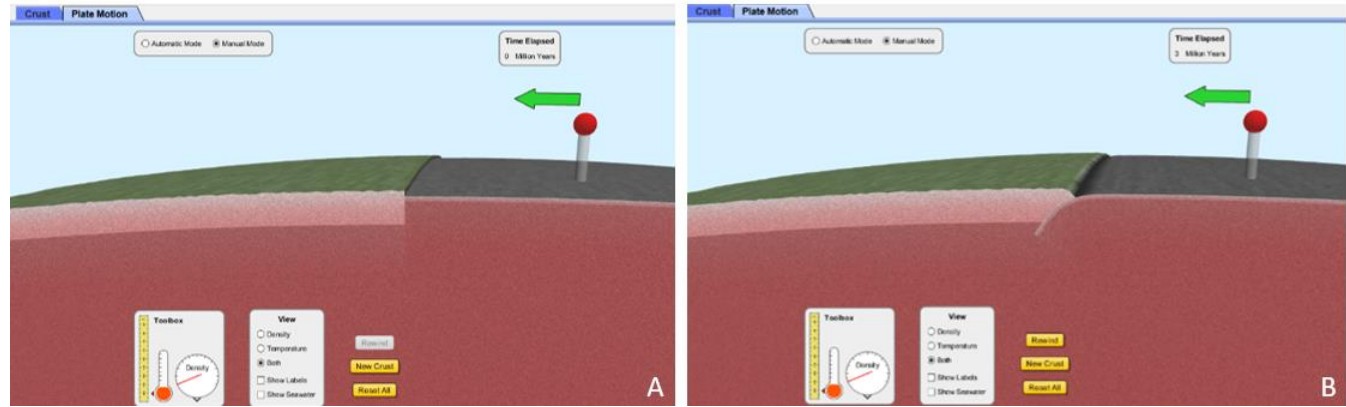

**Figure 4: First image provided to PST that represents a convergent movement with continental and oceanic crust (A). The simulation shows that oceanic crust suffers subduction under continental crust (B) (caption from PhET™ Plate Tectonics).**

Task B1.2. is similar to B1.1., with the difference that the collision is between a plate with continental crust and a plate with recent oceanic crust (new oceanic plate). In the same way, the recent oceanic crust suffers subduction under continental crust.



On task B1.3, PST should simulate a collision between a plate with continental crust and new or old oceanic plates, and compare the subduction angle of the last ones. Using the simulator, they should notice that the angle of subduction on and old oceanic crust is greater than the angle of subduction on a recent oceanic crust (Figure 5)

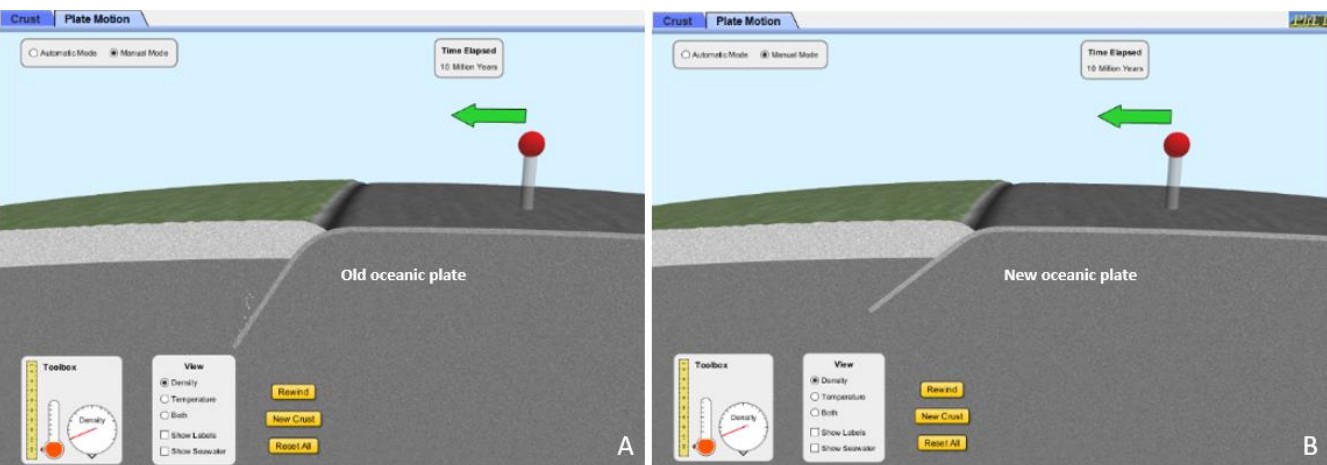

**Figure 5: Subduction angles resulting from the collision between a plate with continental crust and an old oceanic plate (A) and a new oceanic plate (B) (caption from PhET™ Plate Tectonics).**

The last task concerning convergent plate boundaries (task B1.4.) regarded the location of volcanoes that are formed due to the collision between a plate with continental crust and an old or new oceanic plate. Using the simulator, PST should notice that the collision between a plate with continental crust and an old oceanic plate originates volcanoes closer to the continental
margins than the volcanoes resulting from the collision with a new oceanic plate (Figure 6).

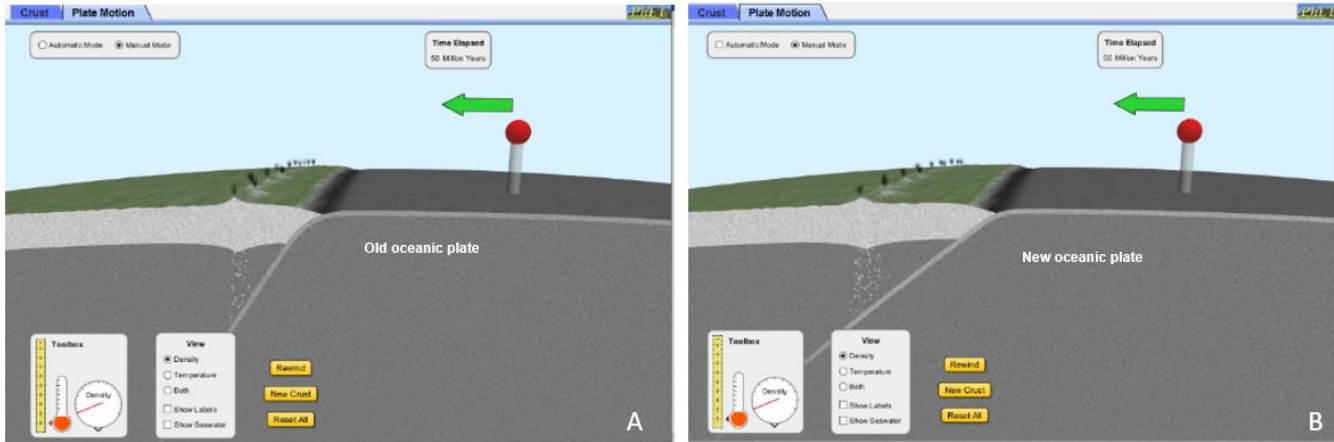

**Figure 6: Location of volcanoes relative to the continental margin resulting from the collision between a plate with continental crust and an old oceanic plate (A) and a new oceanic plate (B) (caption from PhET™ Plate Tectonics).**

The following tasks (B2) regarded the analysis of processes that occur in divergent plate boundaries. On task B2.1., PST must
classify as true or false the following statement: "A rift is only formed due to the divergence of continental crust." To explore this situation, PST could simulate the divergent movements of: continental crust; continental crust and old oceanic crust;



continental crust and new oceanic crust; old oceanic crust; and new oceanic crust. After those simulations, they should notice that the statement is false, because the simulator shows that a rift could also be formed due to the divergence of old oceanic crust (Figure 7).

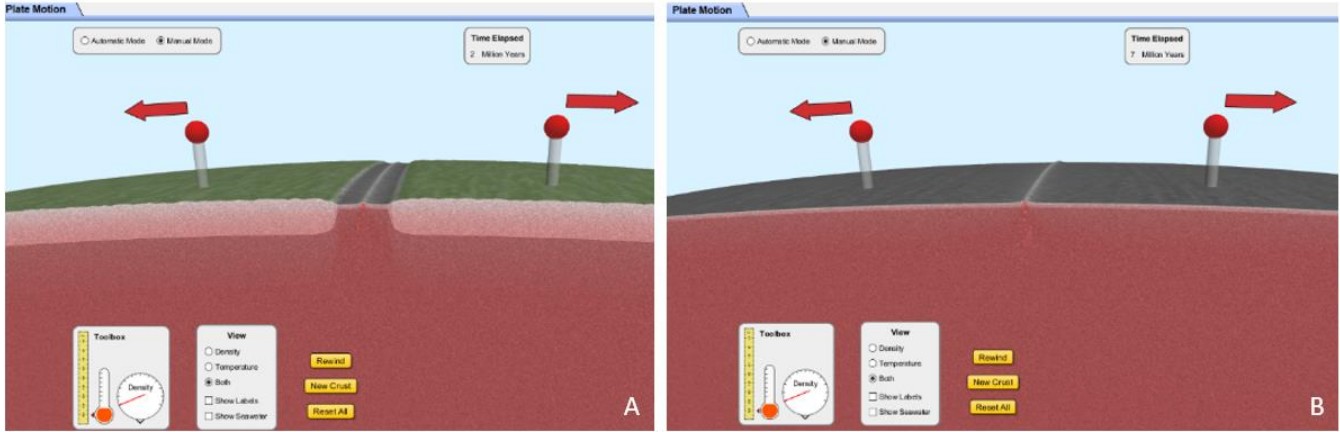


**Figure 7: A rift can be formed through the divergence of the continental crust (A), but also through the divergence of old oceanic crust (B) (caption from PhET™ Plate Tectonics).**

On task B2.2. the image A represented in figure 8 was provided to PST. After observing the tectonic situation represented, PST should select the correct option to complete this statement: "The type of plate movement that causes the formation of

oceanic crust is…". After the manipulation of the different arrows, they should conclude that the correct option is "… divergent movements (red arrows)" as shown in image B (Figure 8).

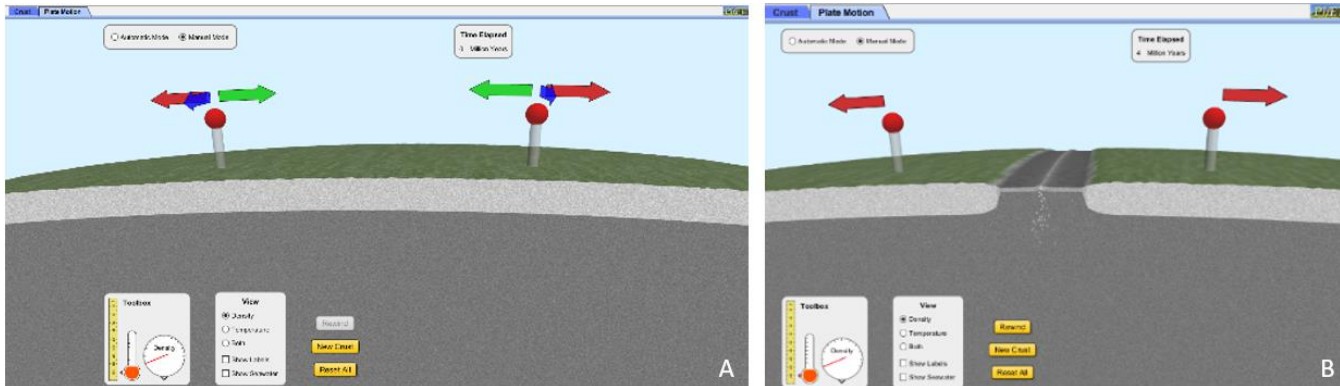

**Figure 8: First tectonic situation provided to PST (A) and formation of oceanic crust due to divergent movements (red arrows; B) (caption from PhET™ Plate Tectonics).**

Table 3 shows PST' tasks B1.1. to B2.2 results.



**Table 3: Preservice teachers' results on tasks B1.1. to B2.2.**

| Task | Frequency of correct answers (n=68) |
|---|---|
| B1.1. Select the process that occurs as a result of the convergent movement between a plate with continental crust and an old oceanic plate. Correct answer: "The old oceanic plate suffers subduction under continental crust." | 100% |
| B1.2. Select the process that occurs as a result of the convergent movement between a plate with continental crust and a new oceanic plate. Correct answer: "The new oceanic plate suffers subduction under continental crust." | 92,6% |
| B1.3. Compare what happens to the subduction angle when a plate with continental crust collides with an old or new oceanic plate. Correct answer: "The collision between a plate with continental crust and an old oceanic plate originates a greater subduction angle than its collision with a new oceanic plate." | 91,2% |
| B1.4. Compare what happens to the location of volcanoes regarding the continental margin when a plate with continental crust collides with an old or new oceanic plate. Correct answer: "The collision of a plate with continental crust and an old oceanic plate originates volcanoes closer to the continental margin." | 94,1% |
| B2.1. A rift is only formed due to divergence of continental crust. Correct answer: False | 64,7% |
| B2.2. The type of plate movement that causes the formation of oceanic crust is… Correct answer: "Divergent movements (red arrows)" | 92,6% |

Results show PST performed very well on all B1 tasks. On B1.1., B1.2. and B1.3. they verified, by using the simulation, that an old or new oceanic plate always suffers subduction in a convergent movement with a continental crust. They also could determine that this phenomenon occurs due to different crust densities. By using the simulator, PST noticed old or new oceanic

plate suffers subduction under continental crust because of its higher density. In addition, PST also observed the subduction angle is different on both convergent plate boundaries (B1.3.) and, consequently, the location of volcanoes on continental margins (B1.4.). B1 tasks have the advantage of moving PST away from the misconception "When two tectonic plates push together, the size, speed, and/or relative position of the plates determines how they interact" (Mills et al., 2017, p. 304).

Performance of PST on task B2.1. was poorer due to the lack of exploring all combinations of plate movements that could

cause a rift formation. Many of the PST groups just simulated the divergence of continental crust, as shown on image A (Figure 8), erroneously concluding it was the only possibility for rift formation. However, this task also had the advantage of moving PST away from the common misconception "When two tectonic plates separate, an empty gap forms" (Mills, Tomas and Lewthwaite, 2017, p. 303), since they could observe that when two tectonic plates separate, a rift is formed.



Concerning B2.2., PST performance was better because they simulated what happened with all movements: divergent
movements (red arrows), convergent movements (green arrows) and transform movements (blues arrows). The simulator
shows that divergent movements (red arrows) are the only ones that can cause the formation of oceanic crust.

### 5.3 Task C. Inquiry about plate tectonics

PST´ problem-solving skills were mobilized on task C, as they were faced with three challenges:

- Task C1. Inquiry about the plate dynamics which should occur to create a non-volcanic mountain range.
- Task C2. Inquiry about the plate dynamics which should occur to create an insular arc.
- Task C3. Inquiry about the plate dynamics which should occur to create a similar process to the one on San Andreas Fault, California.

Table 4 shows the correct answer for each inquiry and the PST results.

**Table 4: Pre-service teachers' C1 to C3 inquiry results.**

| Task | Frequency of correct answers (n=68) |
|---|---|
| C1. Inquiry about the plate dynamics which should occur to create a non-volcanic mountain range | |
| Correct answer: Convergent plate boundaries between continental crusts. | 91,2% |
| C2. Inquiry about the plate dynamics which should occur to create an insular arc. | |
| Correct answer: Convergent plate boundaries between old and new oceanic crust. | 88,2% |
| C3. Inquiry about the plate dynamics which should occur to create a similar process to the one on San Andreas Fault, California. | 97,1% |
| Correct answer: Transform plate boundaries between continental crusts. | |


PST´ performance in the three inquiries was also very good, revealing suitable problem-solving skills. The C1 inquiry shows
PST the process of mountain range formation due to the collision of two plates with continental crust, which turns out to be an
advantage to avoid the following misconceptions: "When two continental tectonic plates push together, both plates are pushed
upward to form volcanoes" (Mills, Tomas and Lewthwaite, 2017, p. 303) and "All mountains are volcanoes" (Mills, Tomas
and Lewthwaite, 2017, p. 304).

The previous tasks (tasks A and B) contribute to the comprehension of major tectonic plate movements and to mobilize
conceptual knowledge about plate tectonics, to carry out the inquiries C1, C2 and C3. Some PST' groups did not answer
correctly to C1 and C2 inquiries due to not exploring all possible combinations of plate movements.

### 5.4 Pre-service teachers' evaluation of the educational resource

The following chart (figure 9) shows the level of satisfaction of a PST sample (19 pairs of PST; P1 to P19) concerning the
contribution of the educational resource to their learning, in a scale of 1 (very unsatisfied) to 10 (very satisfied).



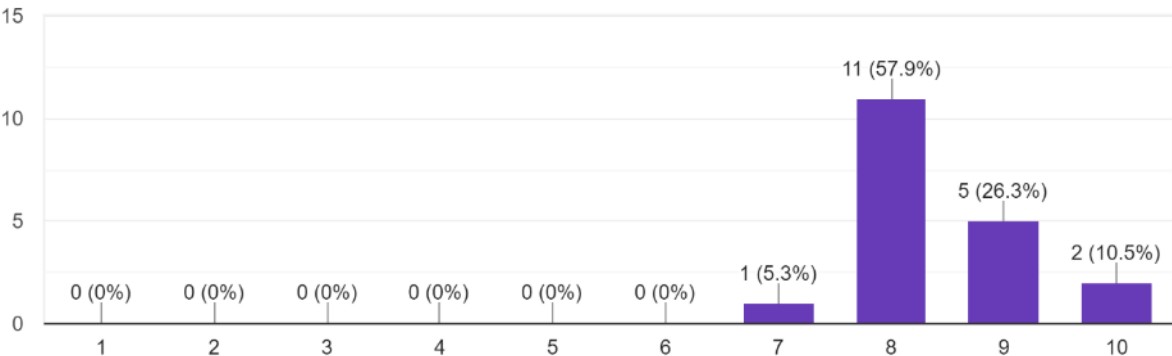

**Figure 9: Chart showing PST' evaluation concerning the contribution of the educational resource CreativeLab_Sci&Math | Plate Tectonics to their learning.**

Results show a high level of satisfaction (≥7) with the educational resource, which is also confirmed by some of the following comments: "This work proposal is very well structured. "(P19); "(..) we really enjoyed doing this work proposal very much." (P17). Table 5 shows the categorization of PST responses regarding the evaluation of the educational resource.

**Table 5. Categorization of PST' answers regarding the contributions of the educational resource to their learning, and improvement suggestions.**

| Category | Subcategory | Frequency (n=19 PST' pairs) |
|---|---|---|
| Contributions to learning | Consolidation of learning regarding plate tectonics resulting from the use of PhET™ plate tectonics simulator | 15 |
| | Consolidation of learning regarding plate tectonics, without specifically mentioning the use of PhET™ Plate tectonics simulator | 3 |
| | PST' collaboration resulting from group work | 2 |
| Improvement suggestions | Absence of improvement aspects | 13 |
| | Instructions for using the simulator | 2 |
| | Other aspects | 4 |

Concerning the contributions to the educational resource to learning, most of PST´ stated PhET™ plate tectonics simulator was a good contribution to "Understanding, through observation, the difference between convergent, divergent and transform boundaries, and the phenomena that happen when plate movements occur." (P3). This is due to the simulator allowing "(…)

observation of the movement and behaviour of different plates, as well as the properties (density, temperature and thickness) of each type of crust (…)" (p15). Some groups emphasized the "(…) interactivity of the simulator (…)" (P11) since it allowed "(…) exploring several hypotheses of transformation according to the data entered." (P17).



Some PST mentioned the educational resource contributed to their learning about Plate Tectonics. However, they did not mention that it was directly due to the use of the simulator. For example, one group considered "It serves to consolidate

knowledge and to better understand the [Plate Tectonic] processes." (P4). Other groups highlighted collaboration between PST as being very important for exchanging ideas about Plate tectonics: "Working in pairs made it easier to understand and allowed the discussion of our ideas." (G6).

Concerning suggestions for improvement, many groups did not present any. The major part of PST was very pleased with the educational resource and its tasks, as shown in the following statements: "We consider the activity was very well elaborated

and there are not any aspects to improve." (P3); "We think the form is very clear and the simulator helps a lot." (p14). However, two of the groups mentioned difficulties with the use of the simulator, suggesting that "Some questions should bring instructions to facilitate the use of the app." (P2). Though, this suggestion goes against the didactical guidelines on the use of PhET™ interactive simulations, which recommend minimal instructions for their use and a free exploration of its content. Other aspects concerned circumstantial situations, as, for example, the work group dynamics: "Since we were working at

distance, there should be a way for each of us to see the form at the same time." (P10).

It was also asked how much time each group spent doing the work. The response range was between 40 and 90 minutes, with an average of approximately 60 minutes.

## 6 Conclusions

As in other works about the advantages of using simulators to promote different skills in students and interest in science (Droui,

2014; Hensberry et al., 2013; Lancaster et al. 2013; McKagan et al., 2008; Nafidi et al., 2018; Perkins et al. 2012; Wieman et al. 2010), the present study highlights the benefits of PhET™ Plate Tectonics interactive simulation to the pre-service teachers (PST) conceptual knowledge about plate tectonics, embedded in a structured educational resource with tasks regarding the characteristics of crust, plate movements, and an inquiry about plate tectonics. PST successfully achieved the learning outcomes that guided the elaboration of this educational resource. In fact, they were capable of: describing the differences

between the oceanic crust and continental crust regarding temperature, density, composition and thickness (Lo1); associating the tectonic plate movements with its geological consequences (Lo2); identifying the tectonic plate movements that cause the formation of some geological structures (Lo3).

Since plate tectonics processes cannot be observed in real time, by using PhET™ Plate Tectonics the PST could observe those processes through the simulation of different movements and crust types. The tectonic processes observed on the simulator

could move PST away from misconceptions about plate tectonics as those identified by Marques & Thompson (2006) and Mills, Tomas and Lewthwaite (2017).



Although the simulator has some limitations, for example, it does not show the mechanism that causes tectonic plate movements, nor processes such as back-arc basin formation, we believe it has a high potential to promote conceptual knowledge about plate tectonics in PST.

In future approaches, it would be interesting to analyse the potential of using simulators such as PhET™ radioactive dating game to explore other geological core ideas, in which students show difficulties in understanding, such as geological time (Dodick & Orion, 2003, 2006).

In short, the CreativeLab_Sci&Math | Plate Tectonics resource allows the improvement of the understanding of some aspects related to tectonic plate movements and encourages the sharing of ideas between PST. The feedback given by PST after
completing the tasks was quite positive, highlighting their engagement with the simulator and its associated tasks. This fact demonstrates the importance of using this DER, such as simulators, to motivate students to learn Geology, as defended by some researchers (Nafidi et al. 2018; Quintana et al., 2004 and Pinto et al., 2014). This DER presents itself as a good pedagogical strategy to be adopted in distance learning, since it can be used autonomously by students in an online context.

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
