# Peer review of "Using PhET™ Interactive Simulation Plate Tectonics on Initial Teacher Education"

_Geoscience Communication, 2020_

## Referee Comment (RC1) · Anonymous Referee #1 · 26 Sep 2020

The article is interesting to disseminate the use of simulators to teach geoscience, namely because there are very few virtual models to be used in geoscience classes. Nevertheless, it is written in an elementary way since the research instruments are very simple and not really explored in the article. The data analysis is too simple: content analysis and descriptive statistical. There is also no reference to the validity and fidelity of the instruments used to gather data. It is necessary to assure it is ethically possible to publish images from a simulator that is not the property of the authors of the study.

As such I considered the study can be published with a minor correction: - at least a better reference and a clarification of the instruments used o collect data - a more precise description of the results, namely with a discussion with reference to the literature in the area - with the certainty that the images con be published.

[Figure]

I consider the article more as a classroom practice description than a research article.

---

## Referee Comment (RC2) · Clara Vasconcelos (Referee) · 13 Oct 2020

I consider the answers of the authors more than satisfactory and I suggest the publication of the article after the incorporation of the comments given by bento and Sara.
* * *

---

## Referee Comment (RC3) · Emer Emily Neenan (Referee) · 13 Nov 2020

Thank you for an interesting paper. The digital resource described in the paper does indeed seem to be useful and this evaluation is important. I see in the discussion that the study was impacted by the COVID-19 pandemic, and I would like to commend the authors for presenting good work despite this considerable setback.

I agree with Reviewer #1 that the discussion of the research instruments is quite elementary. I have seen that the authors have acknowledged this review and added a few sentences, however, I would encourage them to add a little more detail throughout the Methods section of the paper. Section "4.3 Methods of data collection" is fundamental to the paper, but is very short. I would like to know more about the choices of research

instruments, particularly the survey, and also about the methodology of the data analysis. (To prevent the paper becoming longer, I would suggest that perhaps the first part of section 5, concerning the specifics of the tasks, could be trimmed a little shorter without losing vital understanding.)

In places, the language in the paper is grammatically incorrect or confusing. I will give a few examples to the editor to pass on to the authors, but I encourage the authors to have the paper fully proof-read if possible. I understand English is probably not the authors' first language, but I think this paper could be very useful, not only for other geoscience education researchers but also geoscience education practitioners, and clear language will encourage a wider readership.

Overall, I recommend this paper be published subject to minor corrections: 1) adding some more detail to the Methods section, particularly on the methods of data collection and the analysis of the qualitative data 2) improving the clarity of the written English by correcting confusing grammatical errors

Well done again on producing an interesting paper despite the pandemic setback!

---

## Author Comment (AC3) · 27 Nov 2020

We would like to thank referee 2 for the analysis of the preprint version of our manuscript and for the time spent revising the paper. We also sincerely think that the paper will benefit from the incorporation of the suggestions from referee 2. In the following lines, we explain our reply: Referee 2: Overall, I recommend this paper be published after minor corrections: 1) adding some more detail to the Methods section, particularly on the methods of data collection and the analysis of the qualitative data Authors: After reflection, we added a more detailed explanation about the methods of data collection, the sources about design in research education that influenced us and the characterization of the study, as follows: 100. We used an exploratory case study research design (Swain, 2017), because our intent was to achieve first insights

about the contribution of the educational resource CreativeLab_Sci&Math | Plate tectonics to the preservice teachers' learning about plate tectonics. 135. To answer the research question, we used multiple sources of evidence, a defining feature of case studies (Swain, 2017). One was the PST' productions about the educational resource collected through a GForm® questionnaire, mainly with multiple choice questions. The questionnaire was implemented with PST of two Portuguese TTI's in science curricular units in an online teaching context. This digital questionnaire has the advantage of producing an output with the global data of all students' answers. This output was the main instrument of quantitative data collection used. Another method of data collection used was PST' reflections concerning the contribution of the educational resource to their learning, as also the suggestions for its improvement, through a short survey. These reflections were used to collect more qualitative data about PST' learning using the educational resource CreativeLab_Sci&Math | Plate Tectonics. Furthermore, the PST' reflections were also used to enhance the resource. Research teachers course materials were also collected. These materials were used for describing the design and the implementation of the educational resource. Observation of PST' work was also considered, but that method of data collection could not be implemented due to COVID-19 pandemic and the transition to online teaching. 144. At the end of the tasks, a sample of students (19 pairs) was asked to give feedback about the contribution of the educational resource to their learning of plate tectonics, and if they had any suggestions to the improvement of the resource. Through a post-categorization of PST' answers, a qualitative analysis of these data was made using coding categories. During coding, the researchers followed the instruction of Fraenkel et al. (2012). The unit for analysis was PST' sentences (Fraenkel et al., 2012). The coding process emerged two main categories of analysis, "Contributions to learning" and "Improvement suggestions", and three subcategories for each main category. To ensure internal validity, a first analysis made by one of the researchers was followed by a second analysis by the other researcher. When divergences in the categorization process occurred, a discussion was held until a consensus was reached. Extracts of PST' answers were used to

better support the analysis.

Referee 2: Overall, I recommend this paper be published after minor corrections: (...) 2) improving the clarity of the written English by correcting confusing grammatical errors- Authors: We completely understand this suggestion because English is not the authors' first language. We are going to proceed to a fully proof-read of the paper.

We expect that the previous clarifications and additions to the manuscript are according to referee 2' suggestions.

---

## Author Comment (AC4) · 28 Nov 2020

We thank Clara Vasconcelos (referee 1) the final remarks and we inform that we are going to incorporate the comments in the final text.

---

## Author Response (AR1)

**Dear Stephanie Zihms:**

We thank you and the reviewers for your rigorous and constructive review on the preprint version of our manuscript and for the time spent during the thorough revision of the paper.

We have replied to those comments in the following text. Changes to the manuscript related to the editor or reviewers' comments have been highlighted in blue in author's track-changes file. Changes in red are related with the major corrections in the language.

**Referee 1**: "Nevertheless, it is written in an elementary way since the research instruments are very simple and not really explored in the article. The data analysis is too simple: content analysis and descriptive statistical. There is also no reference to the validity and fidelity of the instruments used to gather data. (…) As such I considered the study can be published with a minor correction (…)".

**Referee 1 and Editor:** (…) at least a better reference and a clarification of the instruments used to collect data (…)"

**Authors**: We agree with referee 1 comments about the research instruments. As we stated in the paper, we used an educational resource to gather data about our students' performance. Initially it was our intention to use other research instruments, which was not possible. It is important to say that the educational resource was designed to be used with the physical presence of students and teachers in a classroom in order to allow the collection of field notes. However, COVID-19 pandemics did not allow the implementation of the educational resource in the classroom, which compromised field notes collection and data triangulation. Therefore, after reflection, we added a more detailed explanation about the elaboration and validity of the educational resource, as follows:

121. The educational resource was constantly improved concerning its scientific content, didactic sequence, task's approach and the use of the simulator's potentialities during that implementation, following PST' feedback and teacher's reflections. It was also peer-reviewed by another TTI science education teacher. The internal validity (Cohen et al., 2007; Swain, 2007) of the resource was reinforced by its submission to an open scientific educational resources' repository. During peer-review, the resource was carefully evaluated by geology and other science education university teachers. This process improved the content validity (Cohen et al. , 2007; Fraenkel et al. 2012) of the educational resource, refining its format, the accuracy of the scientific content and questions so that they are clearly understood by the participants, as suggested by Swain (2017), which allowed to provide better explanations sustained by the data (Cohen et al., 2007).

**Referee 1 and Editor**: "(…) a more precise description of the results, namely with a discussion with reference to the literature in the area (…)"

**Authors**: We reinforced the discussion with a thorough comparison of the PST' results with literature in the field.

91. However, additional research to assess the impact of specific simulators on content knowledge is needed (Phuong et al., 2013).

282. B1 tasks have the advantage of moving PST away from common misconceptions about what happens when two tectonic plates push together, e.g. "(…), the size, speed, and/or relative position of the plates determines how they interact", "(…) both plates are pushed upward to form volcanoes" or "(…) for millions of years the larger tectonic plate is pushed upward" (Mills et al., 2017, pp. 303-304).

288. However, this task also had the advantage of moving PST away from common misconceptions about the processes that happen when two tectonic plates separate, e.g. "(…) an empty gap forms" or "(…) loose rock fills the gap that forms between them" (Mills et al., 2017, p. 303) since they could observe that when two tectonic plates separate, a rift is formed.

294. PST' performance achieved through replicating plate movements it's an example of Tan (2007)' idea that simulating reality allows a better analysis and study.

305. PST´ performance in the three inquiries was also very good, revealing suitable problem-solving skills which reinforces the importance of problem-based learning pedagogies (Tan, 2007).

310. Concerning C3 inquiry, PST performance was better when comparing with C1 an C2 tasks results.  The selected example, which addressed to the San Andreas Fault, may have contributed to a better performance by students identifying the correct option, since it is part of the reality that students know because it's a geological subject commonly approached in high schools. This connection to real-world experiences is an important point to take into account in sims exploration (PhET, 2014).

335. These ideas and the results suggest PhET™ Plate Tectonics contributed to the PST' content knowledge about plate tectonics, therefore, adding a contribute to the lack of research about the impact of specific simulators on content knowledge (Phuong et al., 2013). Moreover, these statements are in line with one of the goals of PhET sims, which is to help students to develop and to assess their understanding and reasoning about science topics (PhET, 2014).

353. This statement is important to reflect on, because PhET's Approach to Guided Inquiry (2014) suggests, in point 6, that students should "share their ideas with their partner, working together to answer questions." This process could be committed by the situation described by the student.

**Referee 1**: "(…)  It is necessary to assure it is ethically possible to publish images from a simulator that is not the property of the authors of the study. (...)

**Referee 1 and Editor:** "(…) with the certainty that the images con be published."

In the PhET Interactive simulations online page, in the section "Licensing", it is mentioned that "All simulations available at http://phet.colorado.edu are open educational resources available under the Creative Commons Attribution license (CC-BY). Permission is granted to freely use, share, or redistribute PhET sims under the CC-BY license."

Moreover, in the section "Help Center" there is the following FAQ: "I am a researcher. Do I need a license to use PhET sims and publish research?", whose answer is "No license is needed for research use. Please let us know about your research by completing this form."

To confirm this situation, we sent an email to PhET Help Center, and asked them if we needed a license to use screenshots of PhET sims in the research. The answer was "No license is needed for research use. But you are required to attribute any sims/screenshots you include: PhET Interactive Simulations / University of Colorado Boulder".

179. Figures 1 to 8 represented in this section are screenshots from PhET Interactive Simulations, University of Colorado Boulder.

**Referee 2 and Editor**: "(…) adding some more detail to the Methods section, particularly on the methods of data collection and the analysis of the qualitative data (…)"

**Authors**: After reflection, we added a more detailed explanation about the methods of data collection, the sources about design in research education that influenced us and the characterization of the study, as follows:

105. We used an exploratory case study research design (Swain, 2017), because our intent was to obtain first insights about the contribution of the educational resource CreativeLab_Sci&Math | Plate tectonics to the preservice teachers´ learning about plate tectonics.

153. It was used multiple sources of evidence for answering the research question, a defining feature of case studies (Swain, 2017). The PST´ productions about the educational resource collected through a GForm® questionnaire, mainly with multiple choice questions, was one of those sources. The questionnaire was implemented with PST of two Portuguese TTI's in science curricular units in an online teaching context. This digital questionnaire has the advantage of producing an output with the global data of all students' answers. This output was the main instrument of quantitative data collection used.

Another method of data collection used was PST´ reflections concerning the contribution of the educational resource for their learning, as also the suggestions for its improvement, through a short survey. These reflections were used to collect more qualitative data about PST' learning using the educational resource CreativeLab_Sci&Math | Plate Tectonics. Furthermore, the PST' reflections were also used to enhance the resource.

Research teachers' course materials were also collected. These materials were used for describing the design and the implementation of the educational resource. Observation of PST' work was also considered, but that method of data collection could not be implemented due to COVID-19 pandemic and the transition to online teaching.

169. A sample of students (19 pairs) was asked to give feedback about the contribution of the educational resource to their learning of plate tectonics, and if they had any suggestions to the improvement of the resource, at the end of the tasks. Through a post-categorization of PST' answers, a qualitative analysis of these data was done using coding categories. The researchers followed the instructions of Fraenkel et al. (2012) in the coding process. PST' sentences were the unit of analysis (Fraenkel et al., 2012). From the coding process it emerged two main categories of analysis, "Contributions to learning" and "Improvement suggestions", and three subcategories for each main category. A first analysis done by one of the researchers was followed by a second analysis by the other researcher, to ensure internal validity. When divergences in the categorization process occurred, a discussion was held until a consensus was reached. Extracts of PST´ answers were used to better support the analysis.

Referee 2 and Editor: "(…) improving the clarity of the written English by correcting confusing grammatical errors (…)"

**Authors**: We completely understand this suggestion because English is not the authors' first language. We have proceeded to a fully proof-read of the paper.

**Additional references:**

Cohen, L., Lawrence, M., & Morrison, K.: Research methods in education (6th ed.). Routledge, 2007.

Frankel, J. R., Wallen, N. E., & Hyun, H. H.: How to design and evaluate research in education (8th ed.). McGraw Hill, 2012.

Phong, T. D., Moreland, J. R., Delgado, C., Wilson, K., Wang, x., Zhou, C., & Ice, P.: Effects of 3D virtual simulators in the introductory wind energy course: A tool for teaching engineering concepts. Innovative Teaching, 2(7), doi: 10.2466/04.07.IT.2.7, 2013.

Swain, J.: Designing research education. Concepts and methodologies. SAGE Publications, 2017.

We expect that the previous clarifications and additions to the manuscript are in line with the editor' and referee's suggestions and we remain at your disposal for any necessary clarification or improvement.

---

## Editor Decision (ED1)

Minor corrections checklist

1) adding some more detail to the Methods section, particularly on the methods of data collection and the analysis of the qualitative data

2) improving the clarity of the written English by correcting confusing grammatical errors (see author comments for more information)

3) at least a better reference and a clarification of the instruments used to collect data

4) a more precise description of the results, namely with a discussion with reference to the literature in the area - with the certainty that the images con be published

You addressed these points in your comments to the Authors. Just to make sure they have been completed.

---

## Author Response (AR2)

**Dear Executive Editor Iain Stewart**

**Dear Editor Stephanie Zihms:**

We thank you the final review of our manuscript. We have included all technical corrections and improved the manuscript language. We also added screenshots with higher resolution from PhET™ Interactive Simulations and upload it as a supplement.

We expect that this final improvements in the manuscript are in line with the editors' suggestions and we remain at your disposal for any necessary clarification.

[revised manuscript text omitted]